# Using Graph Representation Learning with Schema Encoders to Measure the Severity of Depressive Symptoms

**Simin Hong**
School of Computing
University of Leeds, UK
scsho@leeds.ac.uk

**Anthony G. Cohn**
School of Computing
University of Leeds, UK
a.g.cohn@leeds.ac.uk

**David C. Hogg**
School of Computing
University of Leeds, UK
d.c.hogg@leeds.ac.uk

## Abstract

Graph neural networks (GNNs) are widely used in regression and classification problems applied to text, in areas such as sentiment analysis and medical decision-making processes. We propose a novel form for node attributes within a GNN based model that captures node-specific embeddings for every word in the vocabulary. This provides a global representation at each node, coupled with node-level updates according to associations among words in a transcript. We demonstrate the efficacy of the approach by augmenting the accuracy of measuring major depressive disorder (MDD). Prior research has sought to make a diagnostic prediction of depression levels from patient data using several modalities, including audio, video, and text. On the DAIC-WOZ benchmark, our method outperforms state-of-art methods by a substantial margin, including those using multiple modalities. Moreover, we also evaluate the performance of our novel model on a Twitter sentiment dataset. We show that our model outperforms a general GNN model by leveraging our novel 2-D node attributes. These results demonstrate the generality of the proposed method.

## 1 Introduction

Deep learning techniques have been frequently used in sentiment analysis (Tan et al., 2013; Mukhtar & Khan, 2018; Liao et al., 2021), particularly in movie recommendations and product reviews and ratings. In the healthcare domain, sentiment analysis with deep learning provides many benefits such as utilizing medical information to increase healthcare quality (Abirami & Askarunisa, 2017; Bi et al., 2020). We focus our work on applications related to mental health such as using deep learning to make a diagnostic prediction.

There is a pressing need to find a convenient and automated method to assess depression severity. Worldwide, more than 300 million people are suffering depression (Organization et al., 2017). Apart from the high prevalence of Major Depressive Disorder (MDD), overall 85% of depressed individuals are underdiagnosed (Falagas et al., 2007). Research has shown that about 30% of patients suffering from an episode of major depression do not seek treatment, with only 10% of them being adequately treated (Falagas et al., 2007). We validate our approach based on its application to the diagnosis of MDD. We demonstrate that our approach can result in more accurate predictions of the severity of depression, which can enhance automated health care decision making.

Medical research (Tsakalidis et al., 2018) shows that text sentiment analysis methods can be effective in making inferences about people's mental states, such as MDD. For example, depressed people use more first person pronouns than non-depressed people because they are more focused on themselves and less connected with others. These may not be reflected in a person's visual appearance, i.e., visual behaviors. Visual data alone will therefore be insufficient for capturing depressive symptoms that are only manifested in a person's verbal utterances.

Mental disorders are difficult to detect. In practice, clinicians first measure the severity of depressive symptoms in patients and then identify depression in them. During in-person clinical interviews, clinicians widely use a structured Patient Health Questionnaire (PHQ) — a clinically validated tool

determining the severity of depressive symptoms across several personal dimensions (Kroenke & Spitzer, 2002; Arseniev-Koehler et al., 2018).

Inspired by the observation of a provided data set of such interviews, we formulated the hypothesis that the context of words in a transcript can be used to generate PHQ scores (Kroenke et al., 2009). This PHQ metric for depressive disorder forms a context constructed with eight items, such as "sleep problem", "anxiety problem", "fatigue problem", "depression problem", and "no motivation or interest in things" (Arseniev-Koehler et al., 2018). We observed that each text (see figure 1) covers information relating to at least one of these eight topics. This motivates use of a deep learning model that represents these eight PHQ topics, thereby introducing an inductive bias into the learning process. Thus the contextual information about these eight PHQ topics should be encoded to determine depression states.

i always feel irritated. i am lazy when i do not sleep well. my mood was just not right, i was always feeling down and depressed and lack of energy. i always want to sleep. i am lack of interest. i have gone to therapy, it has been useful for me in the past. i would love to talk to someone, i just feel like i do not have anyone so i do not depend on anyone. i have always felt depressed in my life, my symptoms were lack of energy, wanting to sleep a lot, lack of interest. my appetite was uncontrollable either lack of or i was just being gluttonous and eating the wrong things. i have notices those changes in my behavior......

Figure 1: An extract from a raw transcript.

We motivate generating graph-level representations for transcripts (as inputs) to encode contextual information from these transcripts (we assume a transcript encodes facts representing depressive symptoms). To achieve this, we propose a novel form for node attributes within a GNN based model that captures node-specific embeddings for every word in the vocabulary. The representations of each word are shared globally and can be updated according to associations among words in a transcript. We summarize the representations of all the words in the transcript to predict depression states.

Intuitively, the generated node-level embeddings maintain records linking certain existing facts that indicate the known symptoms of a subject. When subsequent facts are discovered which indicate additional depressive symptoms, records will be updated by aggregating both old and new facts. In other words, those records will be updated via a message passing mechanism over the transcript context until ideally all major depressive features have been discovered.

In this way, we can represent the most relevant contextual information which is universal across all transcripts – possibly involving the context of eight PHQ topics we discussed above. Using graph structures to capture context-level features is an innovative pathway which we hypothesise can be used to measure different levels of depression. In our experiments, our approach realized a good generalization on a limited, incomplete, and unevenly distributed dataset (see figure 3).

## 1.1 RELATED WORK

Research in measuring the severity of depressive symptoms aims to train a regression model to predict depression scores (Valstar et al., 2016; Alhanai et al., 2018; Tsakalidis et al., 2018). Some prior work (Ringeval et al., 2017; Alhanai et al., 2018; Haque et al., 2018) applies a sequence-level deep learning model to capture implicit depressive signals. Such models, in general, use a multi-modal sentence embedding, rather than a mapping of a whole interview, to predict a PHQ score. Other prior work (Valstar et al., 2016; Cummins et al., 2017) implements statistical functions (e.g., max, min) on short-term features over an entire interview, but this may fail to preserve useful temporal information across an entire interview, such as some short-term signs of regret, anxiety, etc.

Deep learning approaches (Valstar et al., 2016; Alhanai et al., 2018; Song et al., 2018; Du et al., 2019) to fuse multi-modal features in depression detection appear to be particularly promising. Alhanai et al. (2018) proposed a deep model which was trained jointly with the acoustic and linguistic features. Lam et al. (2019) used multi-head attention modules to extract contextual information from clinical text to assess depression tendency. There are existing studies on utilizing deep learning models with pre-trained word embeddings to extract global features from text (Ray et al., 2019; Mallol-Ragolta et al., 2019; Zhang et al., 2020; Solieman & Pustozerov, 2021). The work (Lin et al., 2020; Solieman & Pustozerov, 2021) applied a multi-modal fusion network to summarize all depressive features from both text and audio. They used text features and audio features that were highly correlated with depression severity.

In principle, we should be able to improve performance further using all available modalities, however feature-fusion method may lead to learn a wrong representation encoding inter-subject variability unrelated to depression (Williamson et al., 2016). The feature-fusion approach to learning from small data may cause poor generalization: with a small sample size, the number of features should also be small to avoid the problems of dimensionality and overfitting. However, given that the dimensions of audio and video features are very large, sparse parametrization needs to be taken into account when training machine learning models for learning a joint multi-modal feature vector (Williamson et al., 2016).

## 2 METHOD

We generate a node-level embedding matrix for each word of a transcript. These node attributes represent an underlying relationship between the current word and every other word in the transcript. In this way, a graph neural network (GNN) schematizes underlying associations among words via their internal representations. We propose to aggregate their internal representations to encode information from the context of a transcript generalizing all major depressive symptoms[1].

**Building a Text Graph:** For a given transcript, we build a text graph. We regard all unique words appearing in a transcript as the nodes of the graph. Each edge starts from a word in the transcript and ends with a word within a fixed window on either side of the word.

Let $G = (V, E)$ denote a text level graph. $V$ is a set of nodes representing all the unique words in a given text, and $E$ is a set of undirected edges between pairs of these nodes, each represented by a set of the two nodes at either end of the edge. We build edges by using a fixed size sliding window in a transcript to collect co-occurrence statistics. Each node has an attribute which is a matrix. To emphasise this representational structure of the attribute matrix, we refer to it as a 'schema' (Hammen & Zupan, 1984; Rudolph et al., 1997; Soygüt & Savaşir, 2001; Dozois & Beck, 2008) $U_i \in \mathbb{R}^{n \times d}$. The $j_{th}$ row of $U_i$ is a vector of length $d$ containing the representation that node $v_i$ has of $v_j$; and $n$ denotes the total number of unique words (the vocabulary size) in a corpus.

### 2.1 SCHEMAS

We generate a schema for each word node which performs a role of recording a global context. This global context retains information from interactions between the current word and every other word. In this way, each word node maintains "a dynamic record" (in the form of a schema) of the context from the given transcript. The schemas are progressively updated by a GNN model. This resulting model produces final embeddings of the words in a transcript in relation to all words in the vocabulary, including the other words that make up the current training transcript.

Schemas preserve structures that represent relationships between the identifier word and every other word. Thus we can exploit these schemas to capture context-level features in an explicit way by learning a GNN model. Our innovation is to represent word proximity through the graph structure and co-occurrence within the same transcript within the schema at each node.

---

[1] our code can be found here

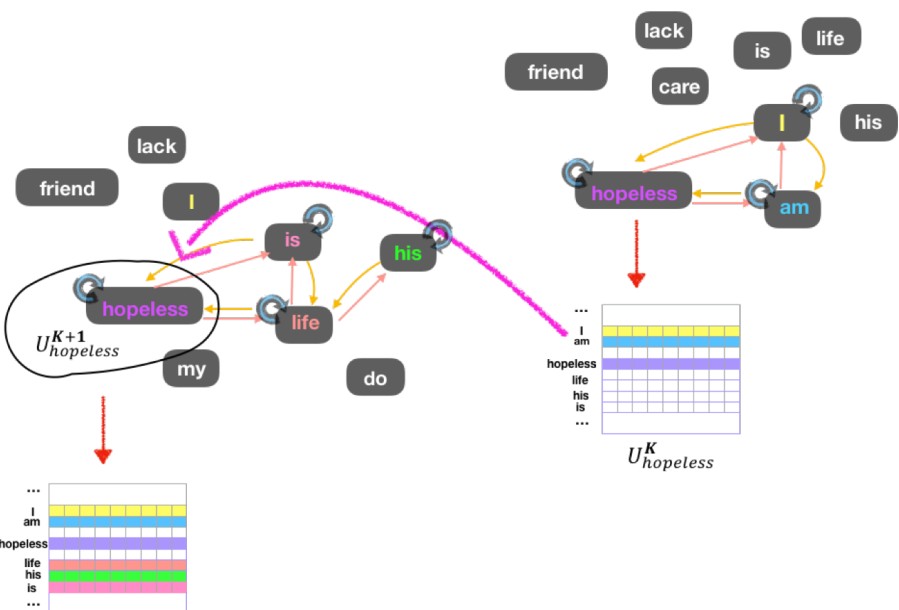

Figure 2: For one of layers of SGNN model $k$, the upper figure showing an output is a schema generated for the word node "hopeless". This schema can be treated as an "inner record" of the node "hopeless", which is formed as $U_{hopeless}^k \in n \times d_k$, where the generated $d_k$-dimensional representation of a node presents as a row. The blank rows correspond to word nodes that have not been encountered with this identifier node "hopeless" in the current text context. The figure on the bottom shows an example of a modified schema of the same word node "hopeless" after learning. Our model using schemas can encode information about associations between the node "hopeless" and its new neighbors, such as a node "his" (colored in green). As a result, the internal representation of this identifier node "hopeless" is updated by the model. Moreover, its original record containing its other already existing neighbors has been explicitly preserved while learning. *Note:* For the convenience of display, in this figure, we set the window size of 1 for displaying associated edges among the nodes; in our actual experiments we use a larger window size.

## 2.2 Schema-based Graph Neural Network (SGNN)

We use multiple passes (layers) of the message passing mechanism (MPM) (Gilmer et al., 2017; Xu et al., 2018) to update the schema at each node of the text graph.

**Initialization:** We first initialise an $n \times d_1$ matrix (in our experiments, we set $d_1 = 32$) as the schema $U_i^{(1)}$ at each node $v_i \in V$. The schema is all zeros apart from the row corresponding to the word associated with this node, which is a random d-dimensional vector using a linear transformation.

**Message passing layer:** In this work, the operation of message passing is split into two steps to update the schema at each node.

We first modify the schema at each node $v_i$:

$$\hat{U}_i^{(k)} = U_i^{(k)}W_1^{(k)} + \frac{1}{n}\mathbf{1}\mathbf{1}^T U_i^{(k)}W_2^{(k)} + \frac{1}{n}\mathbf{1}_i\mathbf{1}^T U_i^{(k)}W_3^{(k)} \quad (1)$$

where $\mathbf{1} \in \mathbb{R}^n$ is a vector of ones.

The first term updates each row independently. The second term operates on the sum of the columns, replicated in each row. The third term operates on the row corresponding to the current word, replicated in each row. These terms are a subset of equivariant linear functions which are computed using the methods of (Maron et al., 2018).

Second we compute the message function, which is defined as:

$$M^{(k)}\left(\hat{U}_i^{(k)}, \hat{U}_j^{(k)}\right) = \hat{U}_j^{(k)} + RELU\left([\hat{U}_i^{(k)}||\hat{U}_j^{(k)}]W_4^{(k)}\right)W_5^{(k)} \tag{2}$$

where $||$ denotes concatenation along the second axis. $\left(W_m^{(k)}\right)_{1 \le m \le 5}$ are learnable parameters. This is essentially a two-layer neural network.

In the next step, each node's schema is updated as:

$$U_i^{(k+1)} = \sum_{j \in N(i)} M^{(k)}\left(\hat{U}_i^{(k)}, \hat{U}_j^{(k)}\right) \in \mathbb{R}^{n \times d_{k+1}} \tag{3}$$

We apply equation (3) as the sum aggregator over the $k$-th layer of the SGNN.

After all $K$ message-passing layers have been applied, the schemas are pooled to get representations of all nodes in the graph. We apply an elementwise max-pooling operater to extract node features. We use the READOUT (Ying et al., 2018; Xu et al., 2018) function which aggregates node features by summing them together:

$$h_G = \sum_{v_i \in V} U_i^{(K)} \tag{4}$$

We use $h_G$ to predict a PHQ score for each transcript. In our experiments, we apply a 2-layer multi-layer perceptron (MLP).

## 2.3 DATASETS

We performed experiments on two datasets, DAIC-WOZ and a twitter sentiment dataset. Whilst our principal interest is in depression prediction, we included the second dataset to show the generality of the method we present here.

- The **DAIC-WOZ** dataset contains video-based facial actions, audio and the conversation transcribed to text for each participant. Both the $6^{th}$ and $7^{th}$ International Audio/Video Emotion Challenge (AVEC) (Valstar et al., 2016) used this dataset. We utilized only the text transcripts of all 142 individuals within the dataset. 43 out of 142 subjects (30%) were labeled as depressed. The provided dataset has been split into a training set having 107 patients and a development set containing 35 patients. In line with prior work (Valstar et al., 2016; Alhanai et al., 2018; Cummins et al., 2017; Haque et al., 2018) and to ensure comparable results, we test on the "development set" from the original competitions (Valstar et al., 2016), since the actual test set is not in the public domain.

  **Privacy:** This data[2] does not contain protected health information(PHI). Personal names, specific dates and locations were removed from the audio recording and transcription by the dataset curators.

- **Borderlands Sentiment Twitter**[3] dataset includes 74682 tweets labeled as positive (28%), negative (30%) and neutral (42%) respectively. Tweets range in length from 2 to 126 words with an average length of 25.3 words per tweet. We extracted 10% of the full data for training.

## 3 EXPERIMENTS

In this section, we show the results generated by our method on two distinct tasks of depression prediction and twitter sentiment classification respectively.

---

[2] https://dcapswoz.ict.usc.edu/ last accessed on 18/09/20

[3] https://www.kaggle.com/cameronwatts/bag-of-words-sentiment-analysis-with-keras/data?scriptVersionId=78350767 last accessed on 11/11/21

Table 1: Comparison of machine learning approaches for measuring the severity of depressive symptoms on the DAIC-WOZ development set using mean absolute error (MAE). The task evaluated is: PHQ score regression. Modalities: A: audio, V: visual, L: linguistic(text), A+V+L: combination. The result marked with a * has been computed by us; the others are taken from the cited papers.

| Regression: PHQ score | | |
|---|---|---|
| Methods | Modalities | PHQ score MAE |
| Baseline Challenge (Valstar et al., 2016) | A+V | 5.52 |
| Gaussian Staircase Regression (Williamson et al., 2016) | A+V+L | 4.18 |
| LSTM (Haque et al., 2018) | A+V+L | 5.18 |
| LSTM (Alhanai et al., 2018) | A+L | 5.1 |
| DCGAN (Yang et al., 2020) | A | 4.63 |
| DepArt-Net (Du et al., 2019) | V | 4.65 |
| LSTM (Alhanai et al., 2018) | L | 5.2 |
| C-CNN (Haque et al., 2018) | L | 6.14 |
| BiLSTM(Lin et al., 2020) | L | 3.88 |
| Multi-level Attention network (Ray et al., 2019) | L | 4.37 |
| *GNN (Gilmer et al., 2017) | L | 4.24 |
| **Our Proposed Approach** | L | **3.76** |

Table 2: Results of ablation studies. We apply 10-fold stratified cross-validation and give mean results. MAE: mean absolute error. RMSE: root mean squared error.

| Metric | MAE | | RMSE | |
|---|---|---|---|---|
| Setting | Train | Test | Train | Test |
| Original(schema-GNN) | 3.85 | 3.52 | 4.48 | 4.32 |
| i) Fast(schema-GNN) | 4.28 | 3.86 | 5.14 | 4.51 |
| ii) Without equivariant linear layers (Maron et al., 2018) | 4.14 | 3.95 | 5.60 | 4.49 |
| iii) Mean Reduction | 4.32 | 5.16 | 4.45 | 5.12 |

## 3.1 THE SEVERITY OF DEPRESSIVE SYMPTOMS PREDICTION

In this experiment, we predict the PHQ score for each participant. The loss objective is mean absolute error (MAE) for regression. We evaluate our SGNN model on DAIC-WOZ benchmark and we then compare our method to the state-of-art works centering on the feature fusion learning algorithms.

We compare our method to prior work on measuring depressive symptom severity. The performance of our method and eleven other methods, including the state of the art method, is set out in Table 1. The results of some models are directly taken from their original papers. Our method outperforms all other methods, despite using only the textual modality.

We found that our model performs better than the standard GNN (Gilmer et al., 2017). The GNN without our schemas utilizes node representations which are initialized by pre-trained 300-dimensional GloVe word embeddings (Mikolov et al., 2013). However, our method constructs schemas which are initialized with random vectors. Moreover the standard GNN propagates node features in terms of vectors while our model uses 2-D node attributes. This change increases the expressive power of the MPM, yielding in better results.

The machine learning algorithms of (Valstar et al., 2016; Williamson et al., 2016) perform modeling statistics using handcrafted features derived from audio, text and (or) visual inputs. However, our method uses just raw text.

We also note that our model performs better than other prior work (Alhanai et al., 2018; Song et al., 2018; Haque et al., 2018; Du et al., 2019; Lin et al., 2020; Ray et al., 2019). That is likely due to the difference of representation learning. Their work uses a multi-modal sentence-level embedding to predict a PHQ score while our model is trained to learn a graph-level embedding of each transcript. The method (Lin et al., 2020) based on the BiLSTM model reaches a similarly low MAE. They apply pre-trained w2v embeddings. However, we train word embeddings with random vectors.

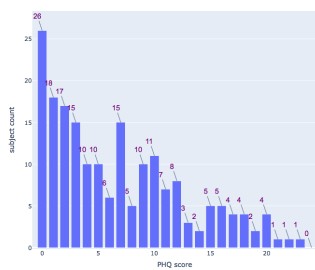

Figure 3: A histogram showing the distribution of PHQ scores on the dataset (of 189 subjects). The X-axis represents a PHQ score ranging from 0 to 24. The Y-axis represents the number of subjects for each score.

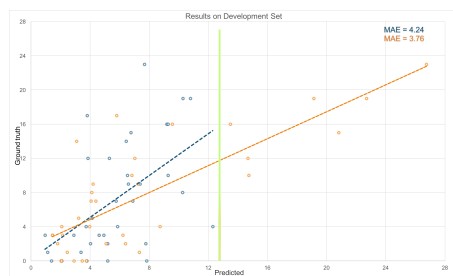

Figure 4: Results on the development set for our graph-based PHQ prediction system, with the true PHQ plotted as a function of predicted PHQ. The red color represents the performance of applying a generic GNN (Gilmer et al., 2017) algorithm, whereas the blue color represents the performance of implementing our SGNN. The green vertical line, as a cut-off, emphasizes a better generalization performance of a small group having scores higher than 13 (on the right of the figure).

### 3.1.1 ABLATION STUDY

To further analyze our model, we perform three ablation studies (see Table 2). To give more reliable measures of performance, we perform 10-fold cross validation on the combined training and development dataset. We concatenate the training and development set as one set and then divide it into 10 folds in a stratified manner. Each time one fold is used for testing and the other 9 folds are used for training.

In (i), we give equations 1-4 (in section 2.2) for 'point-wise' multiplication. According to the results in Table 2, we can see that using convolution layers can better model the relations between words compared with point-wise multiplication layers.

In (ii), we remove the second and third (equivariant) terms in (1). There was a significant reduction in performance (Table 2). As a result, the representation power of our model can increase when parametrized with these layers, thereby improving the model's performance on the data set.

In (iii), we replace the max-pooling operator by a mean-pooling operator. From Table 2, we can see that the result was not good when applying mean operator of the node. Max operator highlighting features can enhance discriminating depressive features, which helps to achieve a better result.

**Qualitative Analysis:** Our model has some incorrect predictions which may be caused by the unequal distribution of dataset (see figure 3). In figure 4, our SGNN model made more biased predictions on those subjects with higher PHQ depression scores (plotted in blue): For instance, the model predicted a PHQ score of around 16 to the patient with an actual PHQ score of 13, and the patient who has the PHQ score of 23 has been overpredicted with a score of around 26. To improve the learning capacity of the model, more labeled training samples with PHQ scores higher than 8 would be needed.

### 3.1.2 INTERPRETABILITY ANALYSIS OF WORD EMBEDDINGS

We give an illustrative visualization of the word embeddings learned by SGNN model. Figure 5 can tell us what the content the model mainly focuses on, therefore helping us to understand its learning process to some extent. We show two examples with predictive depression scores higher than 15. The depression symptoms (of a patient) with corresponding PHQ-scores higher than 14 are classified as moderate-severe or severe by a domain expert (Kroenke et al., 2009).

Compared with the content of the transcripts illustrated by word clouds (figure 5a, figure 5c) of the left-side column, we found that word clouds (figure 5b, figure 5d) on the right-side column represent very different content of the same transcripts. We note that the model can learn to discriminate semantic features relating to depressive symptoms or non-depressive symptoms from the text.

Table 3: Performance of top-10 bi-gram word associates on the DAIC-WOZ development set. The word-word connections, in the context of PHQ-related topics, are generated by the output of the final message passing layer of SGNN model. gt:ground truth

| Transcirpt 1 | Transcript 2 |
|---|---|
| gt score of 16 | gt score of 19 |
| predictive score of 16.46 | predictive score of 17.30 |
| ('married', 'upset') | ('almost', 'thought') |
| ('getting', 'upset') | ('cheated', 'marry') |
| ('anyone', 'argued') | ('unconditional', 'trip') |
| ('family', 'issue') | ('depression', 'psychiatrist') |
| ('feel', 'tough') | ('certainly', 'argent') |
| ('energy', 'explore') | ('exhausted', 'never') |
| ('helping', 'sleep') | ('therapy', 'asleep') |
| ('lack', 'achieve') | ('development', 'issue') |
| ('missing', 'every') | ('married', 'upset') |
| ('sleeping', 'depression') | ('especially', 'breath') |

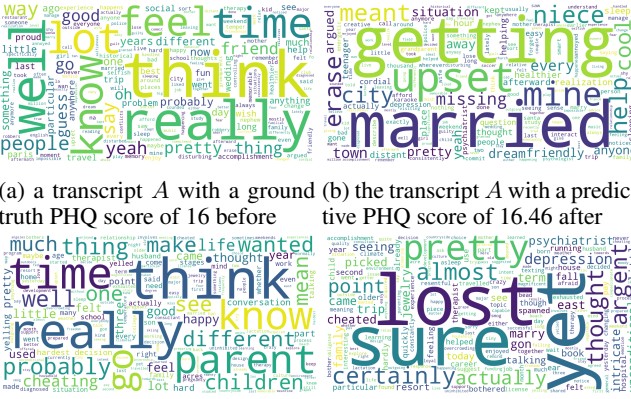

(a) a transcript $A$ with a ground truth PHQ score of 16 before

(b) the transcript $A$ with a predictive PHQ score of 16.46 after

(c) a transcript $B$ with a ground truth PHQ score of 19 before

(d) the transcript $B$ with a predictive PHQ score of 17.30 after

Figure 5: A word cloud depicting words from a transcript on the development set before and after applying SGNN model. The two word clouds on the left-hand column depict the most salient words based on the frequency of their occurrences over two different raw transcripts respectively. The word clouds on the right-hand column illustrate the most focused content selected by SGNN model. The output of the final message passing layer of word embeddings is used. The right and left-hand columns use the same transcript.

**Word Visualization:** We also qualitatively visualize word embeddings, in the form of bi-grams, learned by SGNN model. Table 3 shows the most important word pairs that have been learned to capture context-level semantic features, such as "feel, tough", "therapy, asleep", "sleeping, depression" or "depression, psychiatrist" related to PHQ topics. This demonstrates that the high depression scores are predicted on the basis of appropriate semantic features.

## 3.2 Twitter Sentiment Classification

We evaluate the performance of our SGNN model on classifying three different sentiments on Twitter data (Naseem et al., 2019; 2020; Yenduri et al., 2021; Janjua et al., 2021). We train our model with a total of 7800 tweets. 10-fold stratified cross-validation is also adopted for our experiments. We compare our SGNN model with a general GNN model. Table 4 reports accuracies on both training and testing set. The row one method is initialized with pretrained 300-dimensional GloVe embeddings. The row two method initialize each node with a 2-dimensional (2-D) random node attribute. We observed that our low dimensional 2-D node embeddings can propagate contextual

Table 4: Performance comparison between a general GNN model and the SGNN model with 2-D node attributes on the twitter dataset. We implement a standard GNN without schema encoders. We show the accuracy on a sentiment classification task. We adopt 10-fold stratified cross-validation and report mean with standard deviation in the parentheses for both of these two models. The GNN model is initialized by Glove word embeddings.

| Model | Train | Test |
|---|---|---|
| i) GNN | 0.958 (0.1200) | 0.878 (0.0963) |
| ii) SGNN with 2-D node embedding | 0.951 (0.1146) | 0.941 (0.0013) |

information to the whole text well, while high dimensional pretrained embeddings do not improve classification performances.

Table 4 also shows the performance gains of the proposed model on the Borderlands Sentiment Twitter data when 2-D node embeddings applied. This gain demonstrates that a GNN model with 2-D node attributes can capture more context-level features of sentiments from text, and thus can greatly improve the accuracy of this domain task.

### 3.3 ANALYSIS OF 2-DIMENSION NODE ATTRIBUTES

We compared results of performances between a standard GNN model and our SGNN model on both the depression prediction task and the twitter sentiment classification task using 2-D node attributes.

On the first experiment, our graph representation learning framework using schemas achieves better generalisation on long text. Figure 4 shows that our SGNN model (plotted in blue) generalized the development set much better, particularly for the small group of samples in the class of having scores higher than 13. On the second experiment, we find our SGNN model also performs better on the short text Twitter dataset using schemas, showing that it can also model consecutive and short-distance semantics well.

We suggest that using 2-D node attributes (i.e. schemas) can improve the expressive power of the MPM. By setting a global context (matrices) of each node (as the input), we can obtain some parameters, in an explicit way, that capture the context of a word using each node's schema. On the contrary, using simple 1D node embedding such as GloVe may result in losing contextual information when they are processed by several layers of a learning model.

**Settings:** We set the learning rate as $10^{-3}$, $L2$ weight decay as $10^{-4}$, the dropout rate as 0.5, and the window size as 4 to gather word-word occurrence statistics for both experiments. The loss objective for the first experiment is mean absolute error (MAE); the loss objective for the second experiment is cross-entropy loss. We trained the SGNN for a maximum of 500 epochs using the SGD optimizer (Kingma & Ba, 2014) and stopped training if the validation loss does not decrease for 10 consecutive epochs.

## 4 CONCLUSIONS

Our work shows a novelty of improving the performance of predicting depression states by training a deep graph learning model to learn contextual features from the text. Our results have demonstrated that it is possible to apply deep learning methods to tackle more specific problems within the field, and not just a more general problem, even with limited data. Future work might address finding a way of explaining a GNN model, such as, visualizing the relationship between the underlying depression features and depression scores, which helps us better understand clinical context behind the data.

## 5 ACKNOWLEDGEMENTS

The second and third authors were partially supported by Fellowships from the Alan Turing Institute, UK. The second author was also partially supported by the EU Horizon 2020 Framework under grant agreement 825619 (AI4EU).

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
