# OpenReview forum: "Using Graph Representation Learning with Schema Encoders to Measure the Severity of Depressive Symptoms"
_ICLR.cc/2022/Conference — ICLR 2022 Poster_

### Official Review · Reviewer_ehqq · 2021-10-30

**Correctness:** 4
**Technical Novelty And Significance:** 2
**Empirical Novelty And Significance:** 2
**Recommendation:** 5
**Confidence:** 4

**Main Review:**

Strengths:

-The motivation of the paper is clear. And the overall framework is introduced well. The paper is easy to read.

Weaknesses:

-Graph neural networks have been widely leveraged to improve text analysis. The novelty of the method is limited.

-The experimental analysis is insufficient. Comparisons with recent works on measuring depressive symptom severity are not involved, such as [1], [2], [3].

[1] Muzammel M, Salam H, Othmani A. End-to-End Multimodal Clinical Depression Recognition using Deep Neural Networks: A comparative Analysis[J]. Computer Methods and Programs in Biomedicine, 2021: 106433.

[2] Yang L, Jiang D, Sahli H. Feature augmenting networks for improving depression severity estimation from speech signals[J]. IEEE Access, 2020, 8: 24033-24045.

[3] Lin L, Chen X, Shen Y, et al. Towards Automatic Depression Detection: A BiLSTM/1D CNN-Based Model[J]. Applied Sciences, 2020, 10(23): 8701.


**Summary Of The Paper:**

This paper tries to leverage graph neural networks to improve the performance of measuring the severity of depressive symptoms. Experiments are conducted on a benchmark dataset.

**Summary Of The Review:**

The paper is presented clearly. But the method is not novel. The comparisons with recent works are also not involved.

---

> ### Author Response · Authors · 2021-11-22
> **Response to Reviewer ehqq**
>
> In response to the reviewer’s concerns about a perceived lack of novelty, we would like to explain here the key novel aspects of our paper.
>
> •	A major contribution is to introduce 2-D node embeddings to capture a global representation of each word. We showed our experimental results on a depression prediction task, which outperforms other state-of-art models, using text only. Our work demonstrates the effectiveness of using 2-D node attributes within a GNN based model to encode context-level information. Our proposed method augments the accuracy of the selected domain task. We show superior performance compared to other state-of-art methods with limited text data only.
>
> •	Most research [1-7] focuses on training their model for improving depression severity estimation in terms of processing depression features (from multi-modalities) sequentially. However, we provide a very different perspective of learning this task. We designed our deep learning model based on a graph and we introduced a novel way of using schema structure to capture the context-level information from text. We found that the learning process of our model can be interpretable. Figure 5 (from the revised paper) illustrates that the model can learn to extract context-level features using schema encoders and to predict a depression score based on those learned features.
>
> •	Another novel aspect is that, to the best of our knowledge, this is the first study to use a deep graph-based learning model for depression prediction.
>
> We have updated the paper by adding extra discussion and experiments in the whole body of section 3 of the revised paper. In particular, we have applied the method to a new task, sentiment analysis of a Twitter dataset, which thus contributes further novelty.
>
> We revised the paper based on your concern of lacking an experimental analysis on more recent works. We modified this problem by showing more recent works on depression prediction, which is in section 3 of the paper. Besides, we also added more content about related work in subsection 1.1.
>
> We also thank you for the provided references. We have added all of them in our revised paper.
>
>
> [1] Genevieve Lam, Huang Dongyan, and Weisi Lin. Context-aware deep learning for multi-modal depression detection. In ICASSP 2019-2019 IEEE International Conference on Acoustics, Speech and Signal Processing (ICASSP), pp. 3946–3950. IEEE, 2019.
>
> [2] Muzammel M, Salam H, Othmani A. End-to-End Multimodal Clinical Depression Recognition using Deep Neural Networks: A comparative Analysis[J]. Computer Methods and Programs in Biomedicine, 2021: 106433.
>
> [3] Yang L, Jiang D, Sahli H. Feature augmenting networks for improving depression severity estimation from speech signals[J]. IEEE Access, 2020, 8: 24033-24045.
>
> [4] Lin L, Chen X, Shen Y, et al. Towards Automatic Depression Detection: A BiLSTM/1D CNN-Based Model[J]. Applied Sciences, 2020, 10(23): 8701.
>
> [5] Anupama Ray, Siddharth Kumar, Rutvik Reddy, Prerana Mukherjee, and Ritu Garg. Multi-level attention network using text, audio and video for depression prediction. In Proceedings of the 9th International on Audio/Visual Emotion Challenge and Workshop, pp. 81–88, 2019.
>
> [6] Hanadi Solieman and Evgenii A Pustozerov. The detection of depression using multimodal models based on text and voice quality features. In 2021 IEEE Conference of Russian Young Researchers in Electrical and Electronic Engineering (ElConRus), pp. 1843–1848. IEEE, 2021.
>
> [7] Albert Haque, Michelle Guo, Adam S Miner, and Li Fei-Fei. Measuring depression symptom sever- ity from spoken language and 3d facial expressions. arXiv preprint arXiv:1811.08592, 2018.

---

> > ### Comment · Reviewer_ehqq · 2021-11-26
> > **The revised paper still lacks experimental comparison with recent multi-modal methods.**
> >
> > The revised paper still lacks experimental comparison with recent multi-modal methods, for example [1],[2]. It is not clear whether the method outperforms previous multi-modal works.
> >
> >
> > [1] Muzammel M, Salam H, Othmani A. End-to-End Multimodal Clinical Depression Recognition using Deep Neural Networks: A comparative Analysis[J]. Computer Methods and Programs in Biomedicine, 2021: 106433.
> > [2] Hanadi Solieman and Evgenii A Pustozerov. The detection of depression using multimodal models based on text and voice quality features. In 2021 IEEE Conference of Russian Young Researchers in Electrical and Electronic Engineering (ElConRus), pp. 1843–1848. IEEE, 2021.

---

> > > ### Author Response · Authors · 2021-11-26
> > > **Response to Reviewer ehqq**
> > >
> > > Thank you for your comments. Below you can find our detailed response.
> > >
> > > 1
> > >
> > > The model [1] learns a different text input. The text features used for depression prediction were extracted from both clinical questions and the corresponding patients' responses. If any one of the questions is not referred to during the clinical interview, these methods will not be applicable because they cannot construct a complete feature set. As a conclusion, when applied in evaluating new patient’s conditions, their method may fail. To avoid relying on the clinician’s expertise and to increase the generalization capability, we use the text containing just the patient’s response.  Moreover, they apply a pre-trained w2v embedding, as well as other state-of-art methods apply [3-7]. However, we use random vectors.
> > >
> > > Besides, their work is evaluated using different data distribution: They split the dataset to train, test and validation sets as follows: 146, 18, 18 respectively. On their test evaluation set, their system obtained RMSE = 3.49.
> > > However, we compare our method to state-of-art methods (see table 1 from the revised paper) on measuring depressive symptom severity using the same dataset (details can be found in section 2.3 from the revised paper). The dataset distribution is split into train set (107) and development set (35) respectively. On the development evaluation set, we achieved the result with RMSE = 4.32.
> > >
> > >
> > > 2
> > >
> > > We have not included work that aims to predict a binary prediction of depression (e.g., [2]) rather than a fine-grained prediction of the degree of depression in a PHQ score.
> > >
> > >
> > >
> > > [1] Muzammel M, Salam H, Othmani A. End-to-End Multimodal Clinical Depression Recognition using Deep Neural Networks: A comparative Analysis[J]. Computer Methods and Programs in Biomedicine, 2021: 106433.
> > >
> > >  [2] Hanadi Solieman and Evgenii A Pustozerov. The detection of depression using multimodal models based on text and voice quality features. In 2021 IEEE Conference of Russian Young Researchers in Electrical and Electronic Engineering (ElConRus), pp. 1843–1848. IEEE, 2021.
> > >
> > >  [3] Genevieve Lam, Huang Dongyan, and Weisi Lin. Context-aware deep learning for multi-modal depression detection. In ICASSP 2019-2019 IEEE International Conference on Acoustics, Speech and Signal Processing (ICASSP), pp. 3946–3950. IEEE, 2019.
> > >
> > >  [4] Yang L, Jiang D, Sahli H. Feature augmenting networks for improving depression severity estimation from speech signals[J]. IEEE Access, 2020, 8: 24033-24045.
> > >
> > >  [5] Anupama Ray, Siddharth Kumar, Rutvik Reddy, Prerana Mukherjee, and Ritu Garg. Multi-level attention network using text, audio and video for depression prediction. In Proceedings of the 9th International on Audio/Visual Emotion Challenge and Workshop, pp. 81–88, 2019.
> > >
> > >  [6] Albert Haque, Michelle Guo, Adam S Miner, and Li Fei-Fei. Measuring depression symptom severity from spoken language and 3d facial expressions. arXiv preprint arXiv:1811.08592, 2018.
> > >
> > > [7] Tuka Alhanai, Mohammad Ghassemi, and James Glass. Detecting depression with audio/text sequence modeling of interviews. In Interspeech, pp. 1716–1720, 2018.

---

### Official Review · Reviewer_Bk2m · 2021-10-30

**Correctness:** 4
**Technical Novelty And Significance:** 3
**Empirical Novelty And Significance:** 2
**Recommendation:** 6
**Confidence:** 4

**Main Review:**

+Ves
+ The idea of using schema GNN is interesting for understanding depressive symptoms.
+ The paper is a nice effort with visualizations and nice structure, the ablation studies have been well motivated by pointing out the difference of using different operations.
+ The results indicate the effectiveness of the proposed method.

-Concerns

-Considering the size of dataset, I deem that, the authors should do more experiments on multiple other datasets about depression detection to prove the good performance of the proposed method.
-The related works are insufficient. There are many studies that worked on DAIC-dataset, such as:
1. Mallol-Ragolta A, Zhao Z, Stappen L, et al. A hierarchical attention network-based approach for depression detection from transcribed clinical interviews[J]. 2019.
2. Lam G, Dongyan H, Lin W. Context-aware deep learning for multi-modal depression detection[C]//ICASSP 2019-2019 IEEE International Conference on Acoustics, Speech and Signal Processing (ICASSP). IEEE, 2019: 3946-3950.
3. Ray A, Kumar S, Reddy R, et al. Multi-level attention network using text, audio and video for depression prediction[C]//Proceedings of the 9th International on Audio/Visual Emotion Challenge and Workshop. 2019: 81-88.
4. Zhang Y, Wang Y, Wang X, et al. Text-based Decision Fusion Model for Detecting Depression[C]//2020 2nd Symposium on Signal Processing Systems. 2020: 101-106.
Dinkel H, Wu M, Yu K. Text-based depression detection on sparse data[J]. arXiv preprint arXiv:1904.05154, 2019.
5.Solieman H, Pustozerov E A. The Detection of Depression Using Multimodal Models Based on Text and Voice Quality Features[C]//2021 IEEE Conference of Russian Young Researchers in Electrical and Electronic Engineering (ElConRus). IEEE, 2021: 1843-1848.
6. Lin L, Chen X, Shen Y, et al. Towards Automatic Depression Detection: A BiLSTM/1D CNN-Based Model[J]. Applied Sciences, 2020, 10(23): 8701.
7. Arseniev-Koehler A, Mozgai S, Scherer S. What type of happiness are you looking for? - A closer look at detecting mental health from language[C]//Proceedings of the Fifth Workshop on Computational Linguistics and Clinical Psychology: From Keyboard to Clinic. 2018: 1-12.

-Because the authors said the method could help us understand depression, the paper should contain some interpretability analysis using visual figures or examples of results, which can help us to know which words the model mainly focuses on and the learning process.

-In Figure 2, Is there a threshold to check the neighbor word which is encountered?


Minor comments:

*The authors should add some references about schema structure.

*The reference of the READOUT function should be added.

*The figures are a bit blurry, please replace them with vector diagrams.


**Summary Of The Paper:**

This paper proposes a schema-based GNN method to measure the severity of depression. To gain a global representation of each word, the proposed method constructs word nodes and uses schema structure to capture the context-level information. The main contribution of this paper is the introduction of the schema encoder. The experimental results show the superiority of schema GNN over other SOTA models.

**Summary Of The Review:**

The idea of using schema-based GNN is interesting and have good performance on DAIC dataset. The paper is well written, but many related works are missing. In addition, the authors should add the interpretability analysis.

---

> ### Author Response · Authors · 2021-11-22
> **Response to Reviewer Bk2m**
>
> We thank the reviewer for the positive and constructive comments. We revised the paper by providing extra experiments and discussion which we hope has improved the quality and presentation of the paper. We give a more detailed response as follows.
>
> 1. In order to show that our method works in other domains, we have applied it to a sentiment analysis task using Twitter data. The new subsection 3.2 describing this experiment can be found from the revised version of this paper in detail.
> We trained our SGNN model as a classifier and applied the trained classifier in Twitter sentiment classification, a different domain task from depression severity regression. Our method has achieved an accuracy of 94% of the text dataset over three sentiments: positive; negative; neutral.
> We also compared our method with a general GNN model without schemas on the same Twitter dataset. We adopted 10-fold stratified cross-validation for experiments. We show that our model outperforms a general GNN model by leveraging our novel 2-D node attributes. The experimental results demonstrate the efficiency of our proposed method performing on the same task.  Our SGNN model introducing a novel form for node attributes within graphs thus enables an improvement of the accuracy of two different domain-specific tasks.
>
> 2. Yes, we use a fixed window size (=4 for our experiment) to gather word co-occurrence statistics. This is the way of building edges. We don’t apply any edge attributes to our model, although there are some GNN-based models applying for text classification [1,2] that use point-wise mutual information (PMI) to initialize edge attributes among words.
>
> 3. Thank you so much for the provided references, we have added them all to our revised paper. Moreover, the issue of lacking experimental analysis of more recent work has been addressed. Details can be found both in subsection 1.1 (the related work section) and in subsection 3.1 (the experiments section).
>
> 4. We also added some references introducing this “schema” concept, which is first brought up in the arena of cognitive psychology. We are inspired to design our deep learning model based on graphs by borrowing their heuristic domain findings of recognizing features of depressive symptoms.
>
> 5. A new subsection of interpretability analysis has been added. More details can be found in subsection 3.1.2. We give some visualizations of the model’s outputs performing on two example transcripts from the WAIC-DOZ development set. We observed that the output of the final message passing layer of the SGNN model can be interpretable and understandable in terms of word cloud visualization. Figure 4 can tell us what the content the model mainly focuses on, therefore helping us to understand its learning process to some extent. We also qualitatively visualize word embeddings, in the form of bi-grams, learned by SGNN model. Table 3 can tell us the most important word pairs, such as “feel, tough”, “lose, energi” or “ani diagnos”,  expressing the contextual information of PHQ-related topics, have been learned are used to generate a depression score.  We note that our SGNN model can capture context-level semantic features describing depressive symptoms in an automatic way. We also observed that PHQ-related topics play a significant role for depression prediction.
>
> 6. other minor concerns have been addressed.
>
> [1] Yao, Liang, Chengsheng Mao, and Yuan Luo. "Graph convolutional networks for text classification." Proceedings of the AAAI conference on artificial intelligence. Vol. 33. No. 01. 2019.
>
> [2] Huang, Lianzhe, et al. "Text level graph neural network for text classification." arXiv preprint arXiv:1910.02356. 2019.

---

> > ### Comment · Reviewer_Bk2m · 2021-11-26
> > **Have answered the questions**
> >
> > The authors have added experiments and some explains on concerns. I have improved the score, but I also agree with the third reviewer's opinion, and the overall innovation may be slightly lower for ICLR

---

### Official Review · Reviewer_jCNH · 2021-11-02

**Correctness:** 4
**Technical Novelty And Significance:** 3
**Empirical Novelty And Significance:** 3
**Recommendation:** 8
**Confidence:** 3

**Main Review:**

Strengths:
- Interesting application of GNN in a clinical field where there is a great need of such applications.Clear clinical benefit.
-Shows superior performance compared to prior methods with limited text data only.

Weakness:
- Details of prediction are missing.Is the performance better in patients with severe depression vs mild.While average and aggregated data results are presented, details of underperformance and opportinities for improvement could be identified in subsequent work.
-Unclear why the performance is better compared to multimodal data approaches.Details of the comparison might be worth investigating.

**Summary Of The Paper:**

The authors describe a graph neural network model to predict severity of depression from available text.Application of this method to text data itself is not unique but the clinical context is relevant and important.Many a times in clinical scenarios we do not have access to audio-video data and that's where methods such as these are important utilizing transcribed text.Comparison with prior models is relevant and important and the improvement somewhat significant.

**Summary Of The Review:**

This paper has a very practical clinical context and if generalizable has good opportunity to for application.It is unclear why the performance is superior to other multimodal methods which should be investigated more with details of performance variability.Overall good work.

---

> ### Author Response · Authors · 2021-11-22
> **Response to Reviewer jCNH**
>
> Thank you for your comments and especially for your appreciation of the importance of the work. We also hope that our work will shed light on the development of an efficient and applicable model to assist the domain experts in tackling more complex tasks in their domain areas.
>
> There are some revisions we have made based on your concerns.
>
> To investigate the learning process of our SGNN model, we added more interpretability analysis for a depression prediction task. We use a word cloud and a bi-gram table to interpret the performance of our model. They can be found in subsection 3.1.2 (of the revised paper). We also added one more experiment on a twitter sentiment classification task. More details can be found in the whole body of section 3. We summarize our revisions briefly below.
>
> Visualizing the learning process of SGNN model may help us gaining comprehension about the reason of why our method performances better on recognizing different depressive symptoms than others’ work to some extent.
>
> To achieve this, we use the output of the final message passing layer of SGNN model to visualize its learning process. We show a word cloud visualization  representing the model’s generated results  that can tell us what the content is that the model mainly focuses on for making its prediction. We observed that our model can learn to discriminate semantic features relating to depressive symptoms or non-depressive symptoms from the context of clinical transcripts.
>
> Moreover, we also qualitatively visualize word-word associates with a bi-gram table. Table 3 shows the most important word pairs that have been learned to capture context-level semantic features, such as “feel, tough”, “lose, energi” or “ani diagnos” related to PHQ topics. This demonstrates that the high depression scores are predicted on the basis of appropriate semantic.
>
> We also added extra experiments in our paper which can be found in subsection 3.2.
>
> We compared the performance of our method with a general GNN model on a new dataset: a Twitter sentiment dataset. We included this second dataset to investigate the effectiveness of our novel schema encoders idea. We assume that applying schemas within a graph-based deep learning model may help in improving other sentiment domain tasks on text, i.e., twitter sentiment classification task.
>
> We compared our method with a general GNN model without schemas on the same Twitter dataset. We adopted 10-fold stratified cross-validation for the experiments. We show that our model outperforms a general GNN model on Table 4 (from the revised paper), which demonstrates the generality of the proposed method. We hypothesize that our novel schema encoders help to capture the context-level information which are globally shared over the text. Our model can be trained on different types of text contexts with a good performance.

---

### Author Response · Authors · 2021-11-22
**Dear All Reviewers**

We have uploaded a revised version of the paper which we hope addresses your concerns.

---

### Decision · Program_Chairs · 2022-01-20

**Decision:**

Accept (Poster)

**Comment:**

This paper presents a graph neural network model to predict the severity of depression symptoms from text. It proposes to construct a graph with word nodes and use schema structure to capture the context information in the text. A schema encoder is introduced for modeling the constructed graphs.

Strength:
- Interesting application domain and clear motivation
- Experiment results demonstrate the effectiveness of the method
- Paper writing is clear and easy to follow

Weakness:
- Technical novelty of the method is limited
- Experiment comparison with some recent work is missing
- More in-depth analysis on the method are needed
- Some details of the method pipeline require further elaboration